# Integrated Transcriptome and Metabolome Analysis Reveals Phenylpropanoid Biosynthesis and Phytohormone Signaling Contribute to “*Candidatus* Liberibacter asiaticus” Accumulation in Citrus Fruit Piths (Fluffy Albedo)

**DOI:** 10.3390/ijms232415648

**Published:** 2022-12-09

**Authors:** Xiaoqing Cui, Xuanlin Zhan, Yangyang Liu, Zhenhui Huang, Xiaoling Deng, Zheng Zheng, Meirong Xu

**Affiliations:** Citrus Huanglongbing Research Laboratory, Guangdong Province Key Laboratory of Microbial Signals and Disease Control, South China Agricultural University, Guangzhou 510642, China

**Keywords:** huanglongbing, sequencing, metabolic, titer, fruit

## Abstract

“*Candidatus* Liberibacter asiaticus” (CLas) is a phloem-restricted α-proteobacterium that is associated with citrus huanglongbing (HLB), which is the most destructive disease that affects all varieties of citrus. Although midrib is usually used as a material for CLas detection, we recently found that the bacterium was enriched in fruits, especially in the fruit pith. However, no study has revealed the molecular basis of these two parts in responding to CLas infection. Therefore, we performed transcriptome and UHPLC–MS-based targeted and untargeted metabolomics analyses in order to organize the essential genes and metabolites that are involved. Transcriptome and metabolome characterized 4834 differentially expressed genes (DEGs) and 383 differentially accumulated metabolites (DAMs) between the two materials, wherein 179 DEGs and 44 DAMs were affected by HLB in both of the tissues, involving the pathways of phenylpropanoid biosynthesis, phytohormone signaling transduction, starch and sucrose metabolism, and photosynthesis. Notably, we discovered that the gene expression that is related to beta-glucosidase and endoglucanase was up-regulated in fruits. In addition, defense-related gene expression and metabolite accumulation were significantly down-regulated in infected fruits. Taken together, the decreased amount of jasmonic acid, coupled with the reduced accumulation of phenylpropanoid and the increased proliferation of indole-3-acetic acid, salicylic acid, and abscisic acid, compared to leaf midribs, may contribute largely to the enrichment of CLas in fruit piths, resulting in disorders of photosynthesis and starch and sucrose metabolism.

## 1. Introduction

One of the most damaging citrus diseases worldwide is citrus huanglongbing (HLB), which is mainly caused by the uncultivable phloem-limited α-proteobacterium “*Candidatus* Liberibacter asiaticus” (CLas) [1,2]. HLB is a century-old disease that has been ravaging the citrus industry. However, the axenic culture of CLas has not been established so far. The existing knowledge gap in our understanding of CLas has stymied the development of control methods to mitigate the disease. In addition, the symptoms that are caused by HLB are complicated. The leaves that are infected with CLas show uniform yellowing, blotchy mottling, zinc-deficiency-like, or “green-island” symptoms. The infected fruits are prematurely shed and are small, with curved columella; have aborted seeds, with a bitter taste; and do not color properly [3,4]. These disorders seriously affect the fruit quality and production, thereby threatening the development of the citrus industry worldwide.

Moreover, CLas is unevenly distributed in host plants [5], mainly in the bark tissue, the leaf midribs, the roots, the flowers, and the fruits [6]. The leaf symptom that is most associated with HLB is mottling [7]. Most studies have used the leaf midrib in order to quantitatively detect CLas. However, we recently found that “red-nose” fruit was the most relevant to HLB. Most importantly, the titers of CLas in fruits were the highest, wherein CLas titers in the fruit pith (the fluffy white fibers inside of the peels or outside of the segments) and the central column were extremely high [8,9]. During the fruit pre-ripening and ripening seasons, some studies could use fruit samples to meet the requirement of high CLas titers. Herein, we applied fruit samples as the material in order to study the relationship between the sample type and CLas accumulation.

Transcriptome analysis based on next-generation sequencing (NGS) technology has been widely used in genome-wide studies of plant-pathogen interactions. In recent years, transcriptomic profiling of HLB-affected citrus plants has been studied extensively. Gene expression profiles have been analyzed in citrus plants at different tolerance levels, at different times, and in different tissues after infection by CLas. RNA-seq has been used in order to compare the transcriptional changes in the response of citrus plants to CLas at the early, the medium, and the late stages of infection by graft-inoculation or by vector transmission [10,11]. Regulatory differentially expressed genes (DEGs) that are related to starch, sucrose, and cell wall metabolism were found to be down-regulated in the early stages of infection, while the regulatory genes that are related to salicylate signaling pathways and pathogenesis-related proteins were up-regulated in the late stages of infection [10]. Moreover, hormone signaling, defense response, and secondary metabolism were up-regulated in fruit abscission associated with HLB [11]. Some transcriptomic profiling of CLas-citrus interactions has been performed with susceptible and tolerant citrus species [12,13,14]. Comparatively, the gene expression profiles of the tolerant genotypes were less challenged against CLas infection, especially in starch synthesis and the photosynthesis process. On the contrary, the tolerant varieties exhibited DEGs that were mainly up-regulated with defense-related genes, cell wall metabolism, secondary metabolism, the mitogen-activated protein kinase (MAPK) signaling pathway, and transcription factors. Several studies have analyzed leaves [15,16,17] or fruits [18,19] that were infected with HLB by microarray or transcriptome. They found the gene expression changes in the leaves involved a variety of different processes, including cell wall modification, transport, cellular organization, phytohormones, plant defense, photosynthesis, and carbohydrate metabolism. Most of the HLB-affected DEGs in fruits are classified as transporters or are involved in carbohydrate metabolism and hormone synthesis and signaling. Martinelli et al. [19] compared the DEGs between HLB-affected fruits and leaves. They found distinct expression patterns involving starch biosynthesis, photosynthesis, and sucrose metabolism between the immature and the mature fruits and the leaf samples. A previous study from our group [8] applied transcriptome analysis in order to find the difference in the leaves and the fruit piths both with and without CLas infection. The fruit piths had higher CLas genome depth than the leaf midribs. Compared to the fruit piths, more up-regulated DEGs were identified in the leaf midribs. These DEGs were involved in the biosynthesis of antimicrobial-associated secondary metabolites. Although tens of transcriptome studies have referenced the citrus–CLas interaction, this is the first study on the trancriptomic comparation of leaf and fruit pith in terms of CLas infection. In this study, we further combined the metabolome analyses in order to interpret the relationship between the tissue type and bacterium accumulation.

Plant metabolomics is the profiling of metabolites in tissues, which can be applied as a tool for biomarker discovery. The alterations of metabolite levels that are revealed by metabolomics allow us to understand the final responses of citrus to CLas. The biosynthesis of amino acids and secondary metabolites in citrus leaves were significantly affected by CLas infection, and this host response is different among oranges, mandarins, and grapefruits [20]. Some reports have revealed that there were extreme differences in the metabolites in the leaves of HLB-sensitive and resistant citrus to CLas infection [21,22,23,24,25]. The contents of L-threonine and L-serine in the leaves were significantly up-regulated in leaves of the sensitive varieties, and the tolerant varieties showed higher levels of L-glycine and mannose when they were compared to the sensitive varieties after HLB inoculation [21]. CLas-infected citrus can also alter the long-chain fatty acid, amino acid, and organic acid profiles [21,22,23]. Furthermore, the concentrations of several metabolites, such as phenylalanine, histidine, limonin, and synephrine, were found to be higher in symptomatic fruits [24]. There was also a significant decrease in the concentrations of sugars, proline, and arginine, and an increase in phenylalanine, between orange juice that was collected from CLas-infected citrus trees and that from healthy trees [25]. When comparing different HLB-symptomatic and HLB-asymptomatic tissues of citrus plants, it was found that the symptomatic trees had a higher sucrose content in their leaves but no variation in their roots. However, smaller amounts of proline, betaine, and malate were detected in the HLB-affected symptomatic leaves [26]. These changes in the metabolic processes corroborate the relationship between the CLas levels and the metabolic profiles. Consequently, the generation of the metabolite-based biomarkers of CLas infection or the accumulation in citrus pith in this study contributes to our new understanding of this disease.

Proteome and metabolome analyses of CLas-infected citrus fruits showed that most of the differentially expressed proteins that are involved in glycolysis, the tricarboxylic acid cycle, and amino acid biosynthesis were degraded, and terpene metabolism in the symptomatic fruits was significantly down-regulated [27]. In addition, combined transcriptome and metabolome analyses of CLas-induced fruit pericarp pigmentation showed that different pericarp colors were associated with decreased contents of some carotenoids and phenylpropanoid derivatives, as well as down-regulated proteins of flavonoids, phenylpropanoid derivative biosynthesis pathway proteins, and photosynthesis-antenna proteins [28]. More comprehensively, Chin et al. [29] combined transcriptomics, proteomics, and metabolomics. The results of this study showed that photosynthesis was inhibited, and cell wall modification was activated at the late stages of the infection. Through transcriptome and metabolome analyses of CLas-infected leaves, bark, and roots, Peng et al. [30] found that the immune-related proteins in the CLas-infected leaves were inhibited, and the related genes that are involved in carotenoid biosynthesis and nitrogen metabolism in the roots were down-regulated. However, almost all of the DEGs that are involved in signal transduction and plant–pathogen interaction in the bark were up-regulated. Hence, combined transcriptome and metabolome analysis is further applied here, aiming to gain a better understanding of the specific interactions between the citrus fruit pith and CLas.

Endogenous plant hormones (phytohormones) are the trace organic small molecule compounds that are produced by plants that regulate physiological processes. As the key regulators of plant development and physiological processes, the accumulation or the depression of phytohormones is an intricate mechanism by which the plant adapts to abiotic and biotic stress [31]. Several categories of phytohormones defend plants by promoting specific protective mechanisms against biotic stresses. These phytohormones include auxins (AUX), gibberellins (GAs), cytokinins (CK), abscisic acid (ABA), ethylene (ETH), jasmonates (JAs), and brassinosteroids (BRs). In addition to their unique roles in the different periods of plant growth and development, these hormones can also promote or inhibit each other, and they play individual roles as stress signal triggers [31,32]. Generally, the salicylic acid (SA) and the JA pathways are in an antagonistic relationship with one another [31]. Other hormones, such as ETH, ABA, GAs, AUX, BR, and indole-3-acetic acid (IAA), fine-tune this relationship [32]. To date, there has been no systemic study on the regulation of phytohormones in citrus plants that have been exposed to CLas infection. Understanding phytohormonal homeostasis and signaling in response to CLas in fruits and leaves is essential for understanding citrus performance under pathogen stress.

In conclusion, the unevenly distributed CLas is enriched in the leaf midribs and the fruit piths. Wherein the titer of CLas in the fruit piths is significantly higher than that in the leaf midribs. This study has analyzed the differences between the leaf midribs and the fruit piths from the same branch in order to explore the biological basis of CLas enrichment; however, the uncultivable status of CLas has greatly hampered the research on the reasons and the results of this enrichment. Therefore, in this study, a comprehensive investigation of fruit piths would help us to understand the complex mechanisms between pathogens and host metabolism.

## 2. Results

### 2.1. Phenotype of the CLas-Infected and Uninfected Fruits and Leaves

We first measured the CLas titers of HLB-affected leaves and fruits and confirmed the significant accumulation of this bacterium in the fruits (Figure 1d). Then, the appearance and the shape of the diseased fruits and the healthy fruits were observed. The typical HLB-affected fruits of ‘shatangju’ mandarin in winter had a stem area of the fruit that was already orange, while the stylar end remained green. This was, therefore, figuratively called “red-nose” fruit (Figure 1a). Moreover, the HLB-affected fruits (DFs) were deformed, as shown in the lopsided longitudinal sections with aborted seeds (Figure 1a). The size of the CLas-infected fruits was significantly reduced compared to the healthy fruits in both of the horizontals (4.92 ± 0.28 and 3.89 ± 0.18 for the healthy fruits and the diseased fruits, respectively) and the longitudinal diameter (5.58 ± 0.23 for the healthy fruits and 3.47 ± 0.28 for the diseased fruits) (Figure 1c). The weight of the diseased fruits was also significantly reduced, with 47.27 ± 4.33 g for the healthy single fruit vs. 18.67 ± 5.05 g for the CLas-infected single fruit (Figure 1b). Besides, the sucrose content was influenced significantly by CLas in the fruit piths but not between the infected and the uninfected leaves (Figure 1e).

### 2.2. Quality Control of RNA-seq and UHPLC–MS/MS

Four groups of three-replicated citrus samples, namely DLs, HLs, DFs, and HFs, were collected, and their RNA was extracted for sequencing. Overall, the RNA-seq assay for the 12 samples was qualified for high-throughput analysis at the whole-genome transcriptional level. The base occupancy rates with quality values that were greater than 20 (Q20) and greater than 30 (Q30) for the 12 samples were 97.57~97.87 and 93.05~93.81, respectively (Appendix A). The average total mapping ratio of the 12 samples to the reference genome sequences reached 94.46% ± 0.42% (Appendix A). The intra-group correlation analysis showed that the Pearson’s correlation coefficient between each group of samples of the same treatment was more than 0.912, and that between the different treatments of the same plant part was in the range of 0.8460~0.942. However, there were some differences in the correlation between the samples of the other features. Principal component analysis (PCA) showed that the HLB-affected and the control samples clustered separately. The different groups of samples were divided into four distinct regions in PCA distribution, indicating that the samples in each region had a specific gene profile. All of the biological replicates were located in the same domain (Figure 2a), demonstrating the reliability and the consistency of the transcriptional changes within the replicates.

Similarly, metabolic profiling was qualified for further detailed screening. Four tissue-specific groups were clustered for the metabolites by the partial least squares regression for discriminant analysis (PLS-DA), whereas PC1, PC2, and PC3 explained 50.58%, 10.52%, and 9.27% variances among all of the metabolomics samples, respectively (Figure 2b). The PCA score plots showed that the fruit samples (DF and HF) and the leaf samples (DL and HL) could be distinctly separated, wherein two groups of the fruit samples were separated. The distribution of the two groups of the leaf samples was relatively closer, which actually could be separated by PLS-DA using DAMs (Figure 2c). Similarly, from the heatmap of all of the the detected metabolites, more noticeable differences were observed between the tissue types (Appendix A). A total of 1660 of them could be mapped as known metabolites, including the main lipids, phenylpropanoids, polyketides, organic acids and derivatives fatty acyls, glycerophospholipids, and polyketides (Appendix A), based on HMDB (human metabolome database, http://www.hmdb.ca/ (accessed on 16 March 2021)) and Lipidmaps database (http://www.Lipidmaps.org/ (accessed on 16 March 2021)).

### 2.3. General Description of the Differentially Expressed Genes and Differential Metabolites

The transcriptional expression levels of the genes in the different parts of the citrus plants varied to different degrees. Of the 765 DEGs between the leaf midrib that was infected with CLas and the healthy leaf midrib (DL vs. HL), 466 were up-regulated. A total of 1887 DEGs were identified in the fruit piths that were infected with CLas (DF vs. HF), with 1043 of them being up-regulated (Figure 3a). Only 179 genes were commonly regulated between the fruit pith and the leaf midrib samples after being infected with CLas (Figure 3c). Comparatively, more DEGs were screened between the two tissues (7622 for the healthy group and 6589 for the CLas-infected group). The 4834 DEGs for both HL vs. HF and DL vs. DF were suggested to contribute to the tissue difference, while the 1708 DEGs were specifically differentially regulated in the fruit piths in responding to HLB.

The metabolite intensities of the CLas-infected and the healthy samples between the two tissues were compared. Subsequently, 427 DAMs in the fruits (with 227 of them being up-regulated), and 153 DAMs in the leaves (with 65 of them being up-regulated) could be matched to a corresponding Kyoto encyclopedia of genes and genomes (KEGG) compound ID, suggesting that HLB affected the fruits more extensively than it did the leaves at the metabolic level (Figure 3b). The metabolomics profiles between the different tissues revealed 797 DAMs and 835 DAMs for DL vs. DF and HL vs. HF, respectively. A total of 44 DAMs were affected by HLB in both of the tissues, and 383 DAMs were specifically regulated in the fruit piths after being infected with CLas (Figure 3f). Correspondingly, 451 DAMs were suggested to contribute to the tissue differences, despite the infection status. For the metabolites that had a VIP value of >1.5 and a fold change (FC) of >5 in the corresponding CLas-infected tissues was considered as a putative marker of HLB molecular diagnosis. These included 88 metabolites in the fruits (Appendix A) and 9 metabolites in the leaves (Appendix A), especially the 11 metabolites with fold changes that were larger than 50 (Appendix A) and heteroclitin D that accumulated in both the leaves and the fruits after CLas-infection.

### 2.4. GO and KEGG Enrichment Analyses of Differentially Expressed Genes and Differential Accumulated Metabolites

The gene ontology (GO) classification of the DEGs indicated the significant enrichment of 14, 1, 12, and 17 functional groups in DF vs. HF, DL vs. HL, DL vs. DF, and HL vs. HF, respectively. Most (32/44) of these GO terms were classified as the molecular function (MF). Only four were related to the cellular composition (CC). The other eight were related to the biological processes (BP) (Figure 4). For DL vs. HL, only one MF term that had transcription regulator activity for which all of the 29 DEGs down-regulated was identified, which was also significantly enriched in the other comparisons. The analysis of the CC-related DEGs revealed that they were thylakoid- or photosystem-related (with 12, 26, and 32 in DF vs. HF, DL vs. DF, and HL vs. HF, respectively). Regarding the BPs, most of the DEGs were involved in the cellular carbohydrate biosynthetic process and the pollen–pistil interaction in pollination. In terms of the MF, the most significantly enriched GO terms were combined with heme binding (GO: 0020037) in DF vs. HF and transmembrane transporter activity (GO: 0022857) for DF vs. HF and HL vs. HF. The transferase activity, the transferring glycosyl groups, and the tetrapyrrole binding were all enriched for HL vs. HF and DL vs. DF, with most of the DEGs being up-regulated. For the leaf midrib and the fruit pith that were infected with CLas, the most significantly enriched GO term in the BP category was photosynthesis (GO: 0015979). The GO terms that were specific to the DL vs. DF and HL vs. HF groups included thylakoid related to CC. The sequence-specific DNA binding, the tetrapyrrole binding, and two iron, two sulfur cluster binding were correlated with the BP. In conclusion, the GO-enriched DEGs that are worth further analysis are the transcription regulators in DL vs. HL and those that are related to thylakoid, photosystem, or the cellular carbohydrate biosynthetic process in DF vs. HF, DL vs. DF, and HL vs. HF.

In total, 19 and 15 KEGG pathways were significantly enriched in HL vs. HF and DL vs. DF, respectively (Figure 5a). Of them, nine pathways were common to both of the groups. These tissues-related (the leaf midrib and the fruit pith) pathways included photosynthesis (cic00195), specifically in carbon fixation (cic00710) and antenna proteins (cic00196), biosynthesis of carotenoid (cic00906), tropane, piperidine, and pyridine alkaloid (cic00960), ubiquinone and other terpenoid-quinone (cic00130), metabolism of porphyrin and chlorophyll (cic00860), starch and sucrose (cic00500), and plant hormone signal transduction (cic04075). Three of the pathways, namely photosynthesis, plant hormone signal transduction, and starch and sucrose metabolism, were enriched in three of the four groups of samples. These three pathways were also consistent with the KEGGs of the 4834 DEGs that were commonly differentially expressed in the two tissues. After ‘Shatangju’ was infected with CLas, the pathways of the phenylpropane metabolism, the starch and sucrose metabolism, and photosynthesis of the fruit pith were activated. These three pathways were consistent with the KEGG-enriched pathways of 1708 DEGs regarding the HLB-affected fruit pith. In addition, the 179 commonly enriched DEGs in the CLas-infected leaf midrib and fruit pith were mainly related to the sucrose and starch metabolic pathways (Appendix A). Importantly, the DEGs that were involved in the four pathways, e.g., phenylpropanoid biosynthesis, starch and sucrose metabolism, photosynthesis, and plant hormone signal transduction, will be further explained in detail.

For the DAMs (Figure 5b), 39, 50, 49, and 47 of the pathways were ‘over’-enriched for DF vs. HF, DL vs. DF, DL vs. HL, and HL vs. HF, respectively. Of them, 4, 0, 10, and 1 of the pathways were significantly enriched (*p*-value < 0.05) in the respective groups. The DAM numbers of these pathways were 59, 42, and 7 in DF vs. HF, DL vs. HL, and HL vs. HF, respectively. The significantly enriched pathways in DF vs. HF were zeatin biosynthesis (MAP00908), betalain biosynthesis (MAP00965), the metabolic pathways (MAP01100), and tyrosine metabolism (MAP00350). The MAP01100 pathway was associated with the highest number of metabolites (47 metabolites). For the other pathways, less than 10 DAMs were involved in each of them. Interestingly, all of the DAMs that were involved in MAP00965 and MAP00350 were up-regulated here. The main significantly enriched pathways for DL vs. HL included aminoacyl-tRNA biosynthesis (MAP00970), the biosynthesis of amino acids (MAP01230), and phenylalanine metabolism (MAP00360). All of the DAMs that were related to aminoacyl-tRNA biosynthesis, phenylalanine metabolism, porphyrin and chlorophyll metabolism, glucosinolate biosynthesis, and monobactam biosynthesis were depressed, while the DAMs of the citrate cycle (TCA cycle) were all induced. In HL vs. HF, the only significantly enriched pathway was stilbenoid, diarylheptanoid, and gingerol biosynthesis (MAP00945). Consequently, the pathways warranting further study, according to the KEGG results, include betalain biosynthesis, tyrosine metabolism, phenylalanine metabolism, and porphyrin and chlorophyll metabolism.

Combined with the GO and KEGG enrichment results of the DEGs and the DAMs, the following pathways are particularly worthy of further exploration and discussion: photosynthesis and starch and sucrose metabolism, plant hormone signal transduction, and phenylpropanoid biosynthesis.

### 2.5. Photosynthesis and Starch and Sucrose Metabolism Were Related to the Citrus Response to CLas

The photosynthesis pathway was significantly induced in the leaves (HL vs. HF and DL vs. DF) and in the diseased fruits (DF vs. HF). Unexpectedly, this pathway was not enriched in the leaves in response to the CLas infection. There were 35, 33, and 14 DEGs in HL vs. HF, DL vs. DF, and DF vs. HF, respectively. Notably, 33, 32, and 13 of these were up-regulated in the three comparisons. All of the DEGs in the latter two groups were in HL vs. HF (Figure 6). The fold changes in HL vs. HF were collectively greater than those in the diseased comparison, while the fold changes in DF vs. DF were the lowest among the three groups. Only two of the DEGs were down-regulated in the HL compared to HF, which were not up-regulated in the other two groups. These photosynthesis-related proteins were putatively located in the chloroplast. Most (20 of the 33) of the associated chloroplast proteins were the components of the photosystem I (PSI) subunits. These significantly altered transcripts in the PSI included nine PSI reaction center subunit proteins, five ferredoxins, and three ATP synthase gamma or delta chains. The DEGs that were involved in the PSII included those encoding three PSII reaction center W proteins, three oxygen-evolving enhancer proteins, two PSII proteins of D1, one PSII repair protein, and one PsbP-like protein 1. Although the photosynthesis pathway was not enriched in the DEGs of DL compared to HL, the porphyrin and chlorophyll metabolism pathway was identified by KEGG analysis of the DAMs. The three metabolites that were involved were L-glutamic acid, L-threonine, and biliverdin, and all were decreased by HLB in the leaf samples. When combined with other related studies, these significantly induced DEGs or DAMs could be screened as biomarkers for HLB diagnosis.

Similarly, the starch and sucrose metabolism pathway was enriched in HL vs. HF, DL vs. DF, and DF vs. HF, with 72, 64, and 33 DEGs in each group, respectively (Figure 7a). A total of 92 DEGs were identified in the three datasets. These 92 DEGs were related to starch (12), sucrose (9), fructose (10), hexose (4), glucose (26), glucan (15), trehalose (11), and others (4). Overall, the starch synthetic pathway was induced in the leaves but decreased in the fruits. The genes encoding the starch synthase (SS) and the granule-bound starch synthase (GBSS) that were involved in the starch synthesis pathway had the same expression pattern within the same group. No clear rules could be extracted for the starch degradation process or from the fold change in the DEGs that were influenced by HLB in the fruit. The sucrose synthetic process was depressed, while the sucrose degradation was induced in the leaves compared with that in the fruits, whether they were infected with CLas or not. All of the differentially expressed fructokinase genes were down-regulated in the leaves compared to the related fruits, with two also being depressed in the fruit samples after CLas-infection. A total of 6 alpha-glucosidase genes and 15 beta-glucosidase genes were differentially expressed in at least one group of HL vs. HF, DL vs. DF, and DF vs. HF. Generally, the endoglucanase genes were down-regulated in the leaf samples. In contrast to the patterns that have been mentioned above, five 1, 4-alpha-glucan-branching enzyme genes were significantly induced in DL vs. DF but were depressed in DF vs. HF. Additionally, six genes coding trehalose-phosphate phosphatases and five genes coding alpha, alpha-trehalose-phosphate synthases were all down-regulated in DL compared to that of DF. Hence, from the perspective of different tissues, the starch synthesis in the leaves was induced, as was the sucrose degradation. By contrast, fructokinase, which is an enzyme that converts starch to sucrose and visa versa, was inhibited, indicating that there was a greater propensity for starch accumulation in the leaves.

### 2.6. Plant Hormone Signal Transduction Was Induced upon CLas Infection in the Leaves

The plant hormone signaling transduction pathway, with 106 related DEGs, was enriched in HL vs. HF, DL vs. DF, and DL vs. HL. As can be seen from the diagram, the expression patterns of these DEGs were similar between DL vs. DF and HL vs. HF but were different from those in DL vs. HL (Figure 7b). The two groups also had similar numbers of DEGs, at 78 and 79, of which 54 were both up-regulated or both down-regulated in the two groups. This suggested that the enriched phytohormone signal transduction pathway was mainly related to the difference in the tissue parts. The DEGs in the fruits and the leaves of ‘Shatangju’ involved seven hormones, namely IAA, ABA, ETH, CK, GA, BR, and JA. In general, most of the AUX-related DEGs (29/36) were down-regulated in the leaves. These included 6 down-regulated auxin-responsive protein SAUR genes, 13 auxin-responsive protein IAA genes (with some of these being up-regulated), 3 down-regulated auxin transporter-like protein genes (wherein the gene MSJ143610 was highly depressed in the leaves at 9.04- and 10.43-fold), and 5 down-regulated genes coding indole-3-acetic acid-amido synthetase GH3. Similarly, all of the CK-related DEGs were found to be down-regulated, including seven genes coding two-component response regulator ARRs (ORR9, ARR9, ARR3, ARR4, ARR12, and ARR15) and three genes coding histidine-containing phosphotransfer proteins (HPts). On the contrary, seven of the eight DEGs that were associated with JA were up-regulated. Among the tissue-induced (leaf midrib vs. fruit pith) JA-dependent transcripts was a set of genes encoding TIFY proteins and jasmonic acid-amido synthetase (JAR1). Furthermore, the SA-, GA-, ETH-, and BR-related DEGs exhibited no consistent expression pattern for either DL vs. DF or HL vs. HF. The 21 hormone-related genes in the leaf samples, both before and after infection, were mainly related to ABA, CK, ETH, AUX, IAA, JA, SA, and GA. All of the DEGs that were associated with ABA, CK, and ETH were found to be down-regulated after the leaves were infected with CLas, suggesting that the three signaling pathways were depressed. In summary, we suggest that the JA pathway was induced in the leaves compared to the fruit piths, whereas AUX/IAA- and CK-related signaling pathways were down-regulated. ABA-, CK-, and ETH-associated signaling pathways were depressed in DL vs. HL. Through the analysis of the involved metabolites, we found that CLas significantly influenced the IAA signaling, while the tissue-related phytohormones were JA, GA, SA, and trans-zeatin-riboside (TZR).

Based on this transitional analysis, further expression profiles of the phytohormones were identified. Detectable amounts of the total TZR, BR, and methyl JA were not found in all of the analyzed samples. The phytohormones SA and IAA were detected in the fruit samples, but not in the leaf samples, at 2.88 ± 0.71 ng/g, FW (fresh weight) SA in HF, 7.65 ± 1.97 ng/g, FW SA in DF, 0.77 ± 0.09 ng/g, FW IAA in HF, and 0.35 ± 0.11 ng/g, FW IAA in DF. Gibberellin A1 (GA1) and GA3 were not found in the diseased fruits alone, while the GA4 was present in all of the samples, except the healthy fruits. The other seven hormones, including GA7, JA, jasmonoyl-isoleucine (JA-Ile), ABA, ACC, N6-(Δ2-isopentenyl) adenine (NIA), and N6-(Δ2-isopentenyl) adenosine (NIAS), were present in all of the samples. CLas was associated with the significant enhancement of SA, 1-aminocyclopropanecarboxylic acid (ACC), and GA4, whereas a significant reduction in GA1, IAA, NIA, and NIAS was detected in the fruit piths (Figure 8b). Compared to the CLas-infected leaf midrib samples, CLas infection was related to considerably altered SA, IAA, ABA, and trans-zeatin (TZ) in the diseased fruit piths (Figure 8c). The ABA concentration in the fruit samples was extremely high, at 215.82 ± 73.96 ng/g, FW and 407.84 ± 144.14 ng/g, FW in HF and DF, respectively, compared to those of 5.73 ± 2.32 in HL and 12.04 ± 5.21 in DL. Furthermore, increased amounts of SA, IAA, ABA, NIA, and NIAS were revealed in the healthy fruit piths compared to those in the healthy leaf midribs, as well as significantly decreased total GA4 and ACC contents (Figure 8d). The enrichment of IAA in the fruit piths and JA in the leaf midribs was consistent with the above transcriptome results. By contrast, no significant difference was recorded between the healthy leaf midrib samples and the diseased samples in terms of the levels of all of the phytohormones.

### 2.7. Phenylpropanoid Biosynthesis Was Induced Specifically in the CLas-Infected Leaves

The integrative analysis of the transcriptomics and metabolomics identified a common KEGG pathway in DL vs. HL, namely the phenylpropanoid biosynthesis pathway. A total of five down-regulated DAMs were assigned to the phenylalanine metabolism pathway after the leaves were infected with CLas, which comprised L-tyrosine, phenylacetylglutamine, 2-phenylacetamide, L-phenylalanine, and N-acetyl-L-phenylalanine. L-phenylalanine and L-tyrosine were simultaneously enriched in the other five KEGG pathways. In the ‘phenylpropanoid biosynthesis’ of the CLas-infected leaves, the contents of the chlorogenic acid, scopoline, and 5-O-caffeoylshikimic acid were elevated by 11.8-, 2.3-, and 2.2-fold, respectively in DL vs. HL. Additionally, shikimic acid and phosphoenolpyruvic acid were both induced. The numbers of the DEGs that were involved in this pathway for DL vs. DF, DF vs. HF, and DL vs. HL were 91, 37, and 14, respectively (Appendix A). After being infected with CLas, three beta-glucosidase genes were also up-regulated in the fruit piths. However, in DL vs. HL, there was no enrichment of beta-glucosidase genes. Moreover, in DF vs. HF, a total of thirteen peroxidase genes were enriched, nine of which were up-regulated, while a total of five peroxidase genes were enriched for DL vs. HL, with four of them being up-regulated. In addition, two anthranilate N-methyltransferase genes were enriched in DF vs. HF, and both of them were up-regulated.

Phenylpropanoid biosynthesis was found to be down-regulated in the CLas-infected fruits (DF vs. DL). In the phenylpropanoid metabolism pathway, thirteen shikimate O-hydroxycinnamoyltransferase (HCT) genes and three phenylalanine ammonia-lyase (PAL) genes were enriched in DL vs. DF, most (eleven) of which were up-regulated in the CLas-infected leaves compared to the CLas-infected fruit piths (DL vs. DF). A total of sixteen O-methyltransferase DEGs were enriched in the three groups, namely caffeic acid 3-O-methyltransferase (nine), caffeoyl-CoA O-methyltransferase (four), flavone 3′-O-methyltransferase (two), and one O-methyltransferase domain gene, and most of them were up-regulated in DL vs. DF. In the secondary metabolism of the flavonoids, three 4-coumarate-CoA ligase genes were up-regulated in DL vs. DF. Moreover, eight of the ten cell-wall-degradation-related beta-glucosidase genes were up-regulated.

## 3. Discussion

### 3.1. HLB Symptoms Are Closely Related to the Photosynthetic Reaction of Leaves and Fruits

This study aimed to elucidate the reasons for CLas enrichment in citrus piths compared to leaf midribs of the HLB-susceptible cultivar ‘Shatangju’. Although there have been tens of “-omic” studies on the citrus-CLas interaction, these have mainly used the leaf, the fruit, and the root as the materials. To the best of our knowledge, this is the first study to show the difference between the leaf midrib and the fruit pith by comparing the transcriptional and metabolic profiles. Symptomatically, typical blotchy mottle and zinc-deficiency-like leaves are usually found on HLB-affected trees in the field. The affected fruits are relatively small and they show “red nose” peels and a lopsided shape (Figure 1a), with quality indexes (Figure 1b,c,e) that are consistent with Liang et al. [33]. CLas is unevenly distributed in host plants [5] and it is enriched mainly in the bark, the leaves, the roots, the flowers, and the fruits, particularly in the fruit piths and the central axes [6,8,9]. The CLas titer is also relatively high in the leaf midribs. Due to the convenience of sampling throughout the year, the leaf midribs are commonly used for CLas detection. However, in some cases, seasonal fruit samples could play an important role as special materials. This study has confirmed the accumulation of CLas in the piths of ripe ‘Shatangju’ fruits compared to the leaf midribs and has further explored the biological basis of this phenomenon via combined biomics.

The photosynthesis-related genes are usually enriched in the HLB-affected tissues. However, the expression patterns of this pathway may be converse in the different tissues. For example, most of these genes are down-regulated in the CLas-positive roots and yellowing leaves; however, they are up-regulated in the affected greening fruits [16,34,35]. A total of 33 photosynthesis-related DEGs and 3 related DAMs were found in this study (Figure 6). These include ferredoxin, which acts as an electron receptor in the PSI [35]; ATP synthase gamma, which plays an important role in the regulation of proton motive force in fluctuating light [36]; cytochrome c6 proteins, which provide the electronic connection between the PSI and the PSII reaction centers of oxygenic photosynthesis [37]; and PSII receptor-side proteins (D1 proteins and repair protein PSB27-H1), which function in photodamage [38]. Meanwhile, the activation of the photosynthesis pathway in the fruits after HLB affection is related to the greening symptom that is caused by the disease. While for the leaf samples (DL vs. HL), only a slight (*p* > 0.05) depression of the photosynthesis pathway was detected. This may be due to only midrib samples being collected in this study. However, photosynthesis-related porphyrin and chlorophyll metabolism were enriched in DL vs. HL. In addition to playing an essential role in the processes of photosynthesis, the down-regulated L-glutamic acid, L-threonine, and biliverdin in this pathway were also found contribute to the plant growth [39,40,41]. This might be related to the out-of-season flushing and blossoming of the HLB-affected trees, which remains to be tested scientifically.

### 3.2. CLas Infection Mainly Influences Starch Metabolism in Leaves or Sucrose Metabolism in Fruits

Disrupted photosynthesis can directly impact various biochemical processes in plants, including the carbohydrate balance in the plant tissues [42], which regulates the plant growth and the biomass accumulation [43]. CLas infection stimulates starch accumulation in the host leaf tissues. However, excessive starch accumulation can cause the decomposition of the chloroplasts and the yellowing of the leaves [44], thereby affecting photosynthesis. The starch dynamics are directly related to the differential regulation of the genes that are related to starch degradation and starch synthesis. Fan et al. [45] found that the starch-degradation-related genes were down-regulated after being affected by CLas, leading to the deposition of the starch in the leaves. In this study, the starch biosynthetic-related genes encoding SS, GBSS, and 1,4-alpha-glucan-branching enzyme (GBE) [46] were especially induced by HLB in the leaves, which then contributed physiologically to starch accumulation and phenotypically to blotchy yellow leaves [47]. However, the starch degradation that was related alpha-glucosidase was induced in the fruit piths, suggesting the breakdown of starch into smaller carbohydrates [48]. Hence, from the perspective of the different tissues (leaf vs. fruit), the induction of starch synthesis and sucrose degradation in the leaves indicated a greater propensity for accumulation in the leaves. Consistently, in our study, the sucrose concentrations in the leaves were not significantly affected by CLas. By contrast, no obvious regularity was found either in the starch concentration or in the expression of the genes or the metabolites that are involved in the starch decomposition pathway in these two tissues. Meanwhile, although the sucrose concentration was decreased by HLB, it was still much higher than that in the leaves, which was consistent with the fold-change patterns of the DEGs and the DAMs. The enriched cell wall metabolism and the cellulose degradation related to beta-glucosidase and endoglucanase [49] in the fruit piths indicates an enhancement of cell wall weakening and a reduction in the stress response, which is consequently associated with the enrichment of CLas in this sink tissue.

### 3.3. The Induction of IAA and Depression of JA Are Closely Related to CLas Accumulation in Fruit Piths

A total of 54 plant hormone signaling transduction-pathway-related DEGs were common in DL vs. DF and HL vs. HF (Figure 7b). We further found that 53 of the 54 DEGs differed identically (35 were down-regulated in both of the groups and 18 were up-regulated in both of the groups). This confirmed the reliability of our transcriptome assay, as both of the groups indicated the DEGs in the leaves compared to the fruit piths. Phytohormones play significant roles in plant–pathogen interactions [31]. The CLas-infection-related phytohormones included ABA, CK, ETH, SA, GA, and IAA, while the tissue-related phytohormones were mainly ABA, AUX/IAA, CK, and JA (Figure 7b and Figure 8). IAA is an essential auxin in most plants, and AUX and IAA are critical for plant growth and many of the developmental processes. The relationship between AUX and CK is one of the most-commonly told stories in plant biology, with the levels being inversely correlated in vivo [50]. Cytokinins coordinate some plant developmental processes via the two-component system (TCS) [51]. Almost all of the differentially expressed TCS regulator ARRs and HPts, which act as CK signal transduction regulators [52], were depressed in the fruit piths in this study, which is consistent with the AUX/IAA-related DEGs. Bacteria can use IAA to interact with plants in their colonization strategy and to circumvent the basal plant defense mechanisms [53]. This consequently explains the accumulation status of CLas in the fruit piths. The JA signaling pathway was found to be suppressed based on the transcriptome, metabolome, and target measurements by the high-performance liquid chromatography (HPLC) assay in the fruit piths compared with the leaf midribs in this study (Figure 7b and Figure 8). Jasmonates are a class of polyunsaturated fatty acid-derived phytohormone, while JA is a type of jasmonate. Plant immunity against pathogens is composed of complex mechanism-orchestrated signaling pathways that are regulated by phytohormones e.g., JA [54]. Previous studies have pointed out that JA signaling plays a critical role in CLas infection [11,22,34]. The involved JAR1 was reported to function in sustaining flower development in Arabidopsis and the bacterial blight response in rice [55,56]. The up-regulated (in the leaves) TIFY transcription factors are plant-specific transcriptional regulators. They have been reported to promote plant growth and stress response through JA signaling [57]. Therefore, we suggest that the depression of the JA signaling pathway facilitates the invasion and the colonization of CLas in fruit piths. Different from JA, the abiotic stress signal ABA has dual roles in modulating the plant–pathogen interactions that are partly mediated by cross-talk with JA and SA biotic stress signaling pathways [58]. The function of highly (more than 20-fold, Figure 8b–d) accumulated ABA in fruit piths remains to be explored in future studies.

### 3.4. Depression of Phenylpropanoid Biosynthesis in the Fruit Pith Decreased Defense Responses

Plants defend themselves from pathogen attacks by inducing appropriate defense responses [59]. The phenylpropanoid pathway is related to the production of defense-related compounds [60]. A feature of plant responses to pathogen infection is the enhanced activation of the phenylpropanoid pathway [61]. The phenylpropanoid pathway is up-regulated in the asymptomatic and the early periods of the HLB-infected leaves, whereas it is down-regulated in the later periods of the HLB-infected leaves [62]. In this study, the phenylpropanoid biosynthesis pathway was depressed after CLas infection in the fruit piths compared to that in the leaves by transcriptomic analysis, which is consistent with the metabolomics results. The enriched metabolites included L-phenylalanine, chlorogenic acid, scopolin, 5-O-caffeoylshikimic acid, and L-tyrosine. L-phenylalanine is associated with an increase in water and nutrients, as well as absorption, which can enhance the plants’ photosynthesis, thereby improving the nutritional properties of the plants [63]. In this study, L-phenylalanine was down-regulated, thereby inhibiting photosynthesis. Chlorogenic acid is an antibacterial and antiviral plant extract [64], while scopolin is involved in plant defense responses [65]. The induction of these three metabolites indicates that the defense response to CLas was activated in the leaves.

The ET-induced PAL activity and PAL mRNA were induced by chilling injury in citrus cultivars showing chilling damage, indicating that PAL stands as a defense response of citrus plants against chilling stress [66]. HCTs were reported to be associated with lignin biosynthesis, which is involved in the defense against abiotic and biotic stress [67]. Most of the DEGs encoding these two proteins were up-regulated in the leaves, especially in the CLas-infected leaves. This indicates that the defense response was stronger in the leaves than in the mature fruits, causing the accumulation of this bacterium in the fruit piths. The flavonoids are synthesized through the phenylpropanoid pathway by a set of enzymes. The high flavonoid contents showed high antioxidant capacities, which correlates with increased citrus tolerance to CLas [68]. In this study, we found that a large number of flavonoids were up-regulated in the CLas-infected leaves, presumably significantly improving the antibacterial ability of the leaves and contributing to the high titer of CLas in the fruit piths (dorsal vascular bundle). We also found that the content of the amino acids and the organic acids in the fruits was higher than that in the leaves, and these amino acids and organic acids may help with the reproduction of CLas.

## 4. Materials and Methods

### 4.1. Plant Material

Three six-year-old *Citrus reticulata* Blanco ‘Shatangju’ trees that were evenly affected by HLB were screened from the field of Yangcun, Huizhou city of Guangdong Province in China. All selected trees were in constant growth conditions. Similarly, three CLas-free ‘Shatangju’ trees were selected as the controls. The infected trees were confirmed to be naturally infected in 2018 and the controls were CLas-negative all of the time. A 50-mesh anti-insect screen thoroughly protected each tree with pole and cable frame architecture from the first detection. In addition, the selected trees were exposed to the same natural environment and the same agricultural management. The sampling was carried out after PCR verification of CLas in December of 2020, when the fruits were at the maturing stage. For the fruit samples, 20 fruits with typical HLB symptoms from each infected tree, and 10 normal fruits from each healthy tree, were harvested and pooled as one sample separately. One or two leaves on the same shoot with every collected fruit from the same tree were pooled as one sample. The fruits and leaves were first wiped with a clean wet paper towel using sterilized ddH_2_O. After peeling the fruits, the pith samples were collected using sterilized forceps. The diseased leaf, healthy leaf, diseased fruit, and healthy fruit were marked as DL, HL, DF, and HF, respectively, with three replicates in each group. Each sample was quickly enclosed with aluminum foil and immediately stored in liquid nitrogen. After transferring to lab, the samples were temporarily stored at −80 °C in a freezer prior to further nucleic acid and metabolite extraction. For the leaf samples, petioles of each sample (with ten leaves) were used for DNA extraction, while midribs with some mesophyll tissues of the front parts in the same sample were mixed for RNA extraction. Likewise, the remaining leaf midribs were assigned for metabolomic profiling. All fruit piths in each sample were mixed evenly and assigned for DNA extraction, RNA extraction, and metabolites extraction (Figure 9).

### 4.2. Determination of Phenotypes and Measurement of Carbohydrate Contents

Leaves and fruits were randomly selected from healthy and CLas-infected trees to determine phenotypes and to measure carbohydrate contents. Ten healthy fruits and ten CLas-infected fruits were selected. The fruits were weighed individually. The horizontal diameter and longitudinal diameter of each fruit were measured with a tape measure. Three replications of DL, HL, DF, and HF samples were used for carbohydrate content measurement. To be specific, 100 mg of leaf midribs (or fruit piths) were weighed and assigned for sucrose content determination with a plant sucrose content detection kit (Solarbio Science & Technology, Beijing, China). All data were presented as means ± standard deviation of the mean (Sd). The significance was calculated toward statistical analysis with one-way analysis of variance (ANOVA) using SPSS 13.0 software (* indicates *p* < 0.05, ** indicates *p* < 0.01).

### 4.3. DNA Extraction and PCR Detection of Ca. L. asiaticus

About 100 mg of petioles or fruit piths were chopped into small pieces and lysed by grinding with two silicone beads (3 mm diameter) in 2 mL microcentrifuge tubes with screwcaps. DNA extraction was performed using the E.Z.N.A.^®^ High-Performance Plant DNA Kit (Omega Bio-Tek, Norcross, GA, USA), according to the manufacturer’s instructions. Precipitated DNA was dissolved in nuclease-free ddH_2_O. The concentration of all extracted DNA samples was evaluated using Qubit^®^ 2.0 Fluorometer (Thermo Fisher Scientific Inc., Waltham, MA, USA). For each sample, 100 ng of DNA was used for quantitative real-time polymerase chain reaction (qPCR) using primers based on 16S rDNA of CLas sequences, as described by Bao et al. [69], to identify the CLas quantity. The qPCR amplifications were conducted in a CFX connect real-time system (Bio-Rad, Hercules, CA, USA). The qPCR results were shown as cycle threshold (Ct) values for diagnosis. According to Ct values of the positive control and negative control, the samples with Ct lower than 34 were judged as CLas positive.

### 4.4. RNA Extraction, RNA-seq, and Differential Expression Analysis

The E.Z.N.A.^®^ Plant RNA Kit (Omega Bio-tek, Inc., Norcross, GE, USA) was used to isolate RNA from the liquid-nitrogen-ground tissues of 12 samples (Figure 9). After extraction, the quality of the total RNA samples was spectrophotometrically verified with fluorescence spectroscopy (Qubit^®^ 2.0 Fluorometer) and Agilent 2100 (Agilent Technologies Inc., Santa Clara, CA, USA). The detection results showed that the quality of the 12 RNA samples could meet the requirements of transcriptome sequencing (Appendix A). High-throughput sequencing was carried out on an Illumina HiSeq3000 system with 150 bp paired-end reads by a commercial sequencing company. FastQC tools were applied to assess the quality of the raw data. Clean reads were obtained by removing reads containing adapters, ploy-N reads, and low-quality reads from the raw data.

First of all, paired-end clean reads were aligned to the reference genome “*Citrus reticulata*” using Hisat2 v2.0.5. The mapped reads of each sample were assembled with StringTie (v1.3.3b) [70] in a reference-based approach. A total of 14,712 new transcripts were obtained, including 2323 newly predicted genes. The number of reads mapped to each gene were counted using featureCounts v1.5.0-p3. The FPKM (fragments per kilobase of transcript per million fragments mapped) of each gene was calculated based on the length of the gene and the read count mapped to it. PCA was then performed based on the FPKM of all samples. Moreover, differential expression analysis was performed using the DESeq2 R package (1.20.0). The resulting *p*-values were adjusted using Benjamini and Hochberg’s approach to control the false discovery rate. The genes with an adjusted *p*-value of ≤0.05 and |log_2_Fold change| of ≥1 were assigned as differentially expressed. Eventually, GO and KEGG pathway enrichment analyses of DEGs were implemented by the cluster Profiler R package for systematic analysis of gene functions, in which the gene length bias was corrected.

### 4.5. Metabolite Analysis by UHPLC–MS

The tissues (100 mg) of each sample were individually treated for ultra-high performance liquid chromatography–tandem mass spectrometry (UHPLC–MS/MS) analyses for quantifying metabolites [71]. UHPLC–MS/MS analyses were performed using a Vanquish UHPLC system (Thermo Fisher, Hannover, Germany) coupled with an Orbitrap Q Exactive^TM^ HF mass spectrometer (Thermo Fisher, Hannover, Germany) in Novogene Co., Ltd. (Beijing, China). The raw data files generated by UHPLC–MS/MS were processed using the Compound Discoverer 3.1 software to perform peak alignment, peak picking, and quantitation for each metabolite. Then, the peaks were matched with the mzCloud, mzVault, and MassList databases to obtain accurate qualitative and relative quantitative results. The metabolic alterations among the four experimental groups were visualized using the PCA via partial least-squares-discriminant analysis at metaX [72]. The statistical analysis was performed using the software of R (version R-3.4.3), Python (Python 2.7.6 version), and CentOS (CentOS release 6.6). We applied univariate analysis (*t*-test) to calculate the statistical significance (*p*-value). The metabolites with VIP of >1 and *p*-value of <0.05 and FC of ≥2 or FC of ≤0.5 were considered to be differential metabolites. The metabolites of interest were filtered based on log_2_(FC) and -log_10_(*p*-value) of metabolites by ggplot2 in R language. The data were normalized using z-scores of the intensity areas of differential metabolites and were plotted with Pheatmap package in R language. Python-3.5.0 and R-3.4.3 were used to analyze and visualize the correlation of the transcriptomic and metabolomic data and to screen the common KEGG pathways between them.

### 4.6. Phytohormone Profiling

A total of 100 mg of each of the 12 samples was used for the phytohormone profile (Figure 9). The tissue samples were weighed after being pulverized with a homogenizer at 25 out of 30 for 20 s. Specifically, the power was added to a cold 50% acetonitrile (ACN) solution and treated with ultrasonic for 3 min followed by 30 min incubation under 4 °C. After centrifugation at 12,000 rpm for 10 min, the supernatants were collected into a solid-phase extraction cartridge and mixed with 1 mL 100% MeOH and 1 mL ddH_2_O. After centrifugation, 50% and 30% ACN solutions were used to rinse the column once. The solution was dried with nitrogen and dissolved in 200 µL 30% ACN.

The contents of the total endogenous stress hormones in the individual samples were detected. The assessed endogenous stress hormones of IAA, GA1, GA3, GA4, GA7, JA, JA-Ile, SA, and ABA were analyzed in the negative scan mode as [M−H]−ions. However, TZ, TZR, NIA, NIAS, BR, 1-aminocyclopropanecarboxylic acid (ACC), and methyl jasmonat were analyzed in the positive scan mode as [M + H]+ ions. Standards and appropriate precursor-to-product ion transitions representing a major fragmentation path unique to each phytohormone were identified and prepared (Appendix A). The UPLC (Vanquish, UPLC, Thermo, Waltham, MA, USA) and high-resolution mass spectrometry (Q Exactive, Themo, Waltham, MA, USA) were applied to collect the hormone data. The resulting data were processed with TraceFinder software.

## Figures and Tables

**Figure 1 ijms-23-15648-f001:**
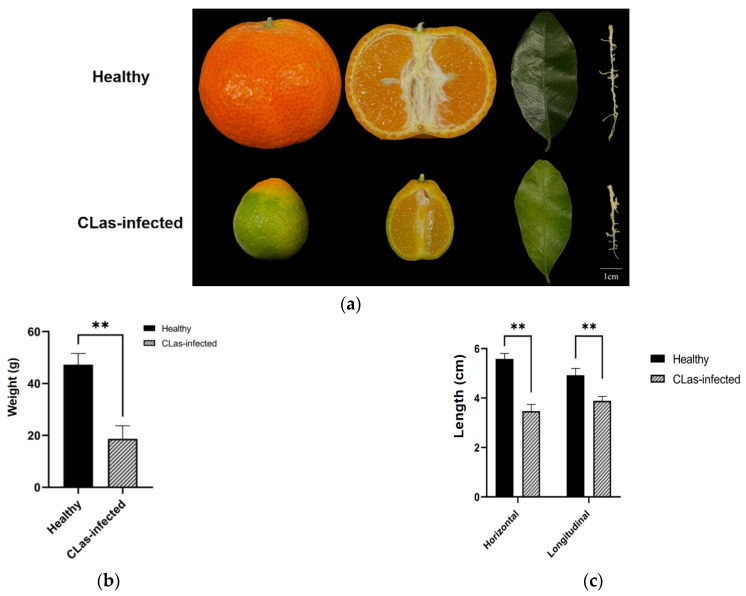
Huanglongbing (HLB) symptomatology and characteristics on the leaf and fruit samples of *Citrus reticulata* Blanco ‘Shatangju’ affected by HLB. (**a**) Healthy (up) and CLas-infected (down) fruit samples. (**b**) The average weight of the fruits. (**c**) The average diameter of the fruits. (**d**) Quantitation of CLas in the leaf and fruit samples. (**e**) Sucrose content (mg/g fresh weight) of the leaf and fruit samples. HLs indicate leaf samples of healthy ‘Shatangju’. HFs are fruit samples of healthy ‘Shatangju’. DLs are leaf samples of ‘Shatangju’ with HLB. DFs are fruits samples of ‘Shatangju’ with HLB. All data are presented as means ± standard deviation of the mean (Sd). The data were subjected to statistical analysis by one-way analysis of variance (ANOVA) using SPSS 13.0 software (** indicates *p* < 0.01, ns indicates there is no significant difference).

**Figure 2 ijms-23-15648-f002:**
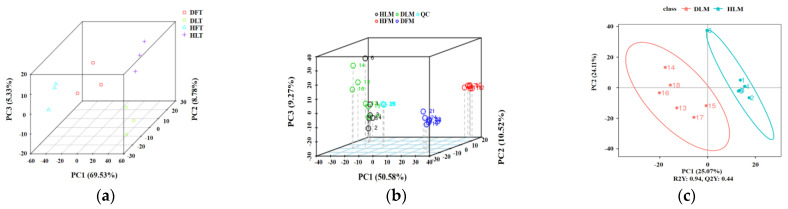
Principal component analysis (PCA) plots for RNA-seq data and mass spectrum data of *Citrus reticulata* Blanco ‘Shatangju’ samples and the quality control of the samples. (**a**) The PCA 3D plot of the transcriptomic (T) on four groups of samples. (**b**) The PCA 3D plot on four groups of samples based on metabolomics (M) data. (**c**) The PLS-DA score plot of two groups of leaf samples based on metabolomics (M) data. The abscissa PC1 and the ordinate PC2 represent the scores of the first and second principal components, respectively. The scattered points of different colors represent the samples of the different experimental groups. DF, diseased fruit pith. DL, diseased leaf. HF, healthy fruit pith. HL, healthy leaf. T, transcriptome analysis. M, metabolome analysis.

**Figure 3 ijms-23-15648-f003:**
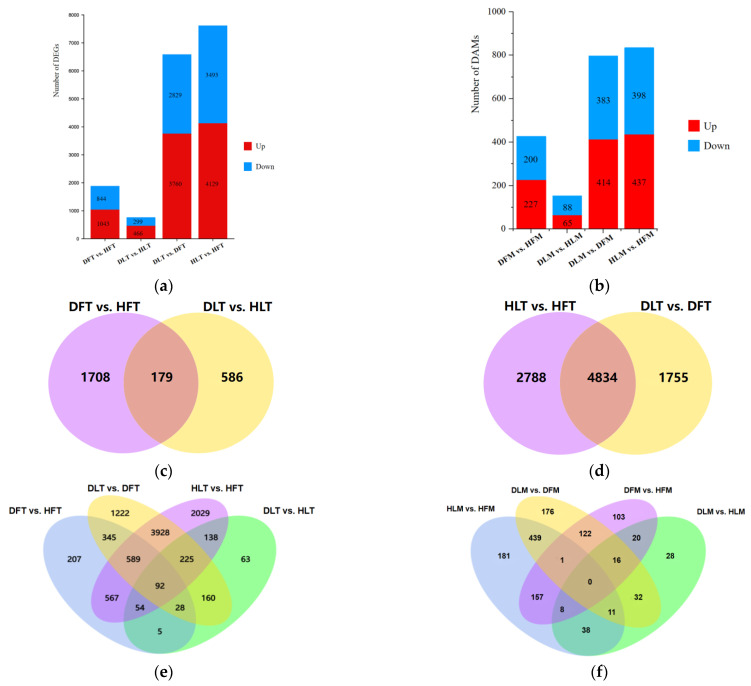
The histogram and Venn diagrams of differentially expressed genes (DEGs) and differentially accumulated metabolites (DAMs) in huanglongbing (HLB)-affected and healthy leaf and fruit pith samples. (**a**) The histogram of the number of DEGs. (**b**) The histogram of the number of DAMs. (**c**–**e**) The Venn maps of DEGs in different combinations. (**f**) The Venn diagrams of DAMs of CLas-infected and healthy tissues. DF, diseased fruit pith. DL, diseased leaf. HF, healthy fruit pith. HL, healthy leaf. T, transcriptome analysis. M, metabolome analysis. Different colors indicate different comparison combinations.

**Figure 4 ijms-23-15648-f004:**
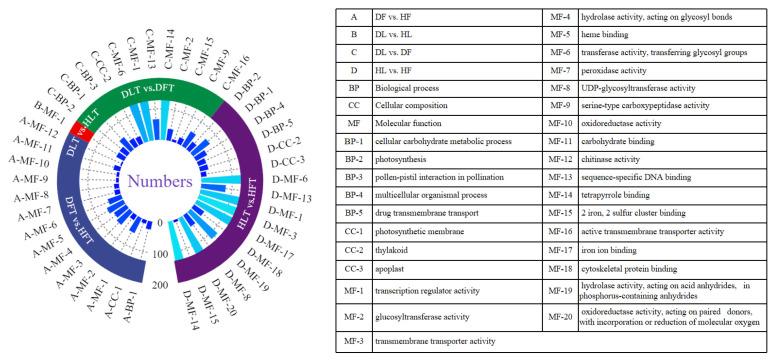
Ring diagram of GO enrichment analysis of the differentially expressed genes of *Citrus reticulata* Blanco ‘Shatangju’ in response to “*Candidatus* Liberibacter asiaticus” infection. In the figure, the different colors represent different functional classifications. DF, diseased fruit pith. DL, diseased leaf. HF, healthy fruit pith. HL, healthy leaf. T, transcriptome analysis. M, metabolome analysis. Different colors on the ring mean different comparison groups. The density of the blue-color bars inside the ring differentiates the number of DEGs enriched in the specific GO terms.

**Figure 5 ijms-23-15648-f005:**
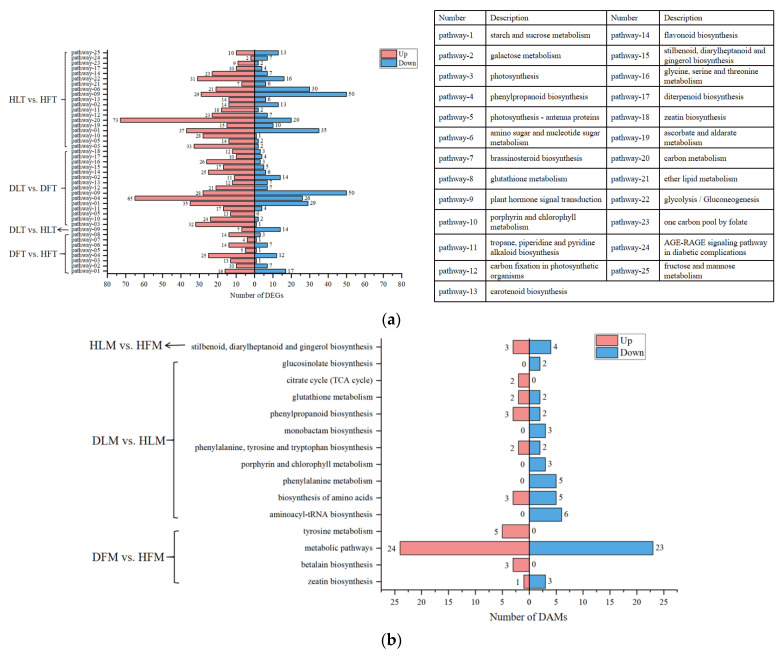
The histogram of KEGG enrichment analysis by differentially expressed genes (DEGs) and differential accumulated metabolites (DAMs) of *Citrus reticulata* Blanco ‘Shatangju’ in responding to “*Candidatus* Liberibacter asiaticus” infection. (**a**) The histogram of KEGG enrichment analysis in DEGs. (**b**) The histogram of KEGG enrichment analysis in DAMs. The horizontal coordinate is the number of DEGs and DAMs enriched in the KEGG pathway, and the vertical coordinate is the KEGG pathway. The number bar of up-regulated genes and metabolites histogram in KEGG nodes is red. The number bar of down-regulated genes and metabolites histogram in KEGG nodes is blue.

**Figure 6 ijms-23-15648-f006:**
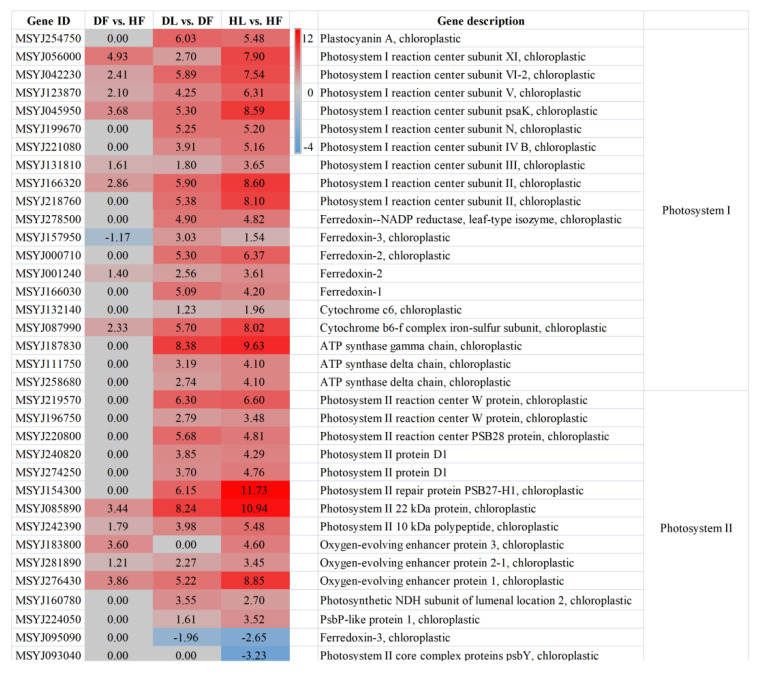
Log_2_Fold change cluster diagram of the differentially expressed genes in the photosynthesis pathway. DF, diseased fruit pith. DL, diseased leaf. HF, healthy fruit pith. HL, healthy leaf. The color bar represents a range of Log_2_Fold change values.

**Figure 7 ijms-23-15648-f007:**
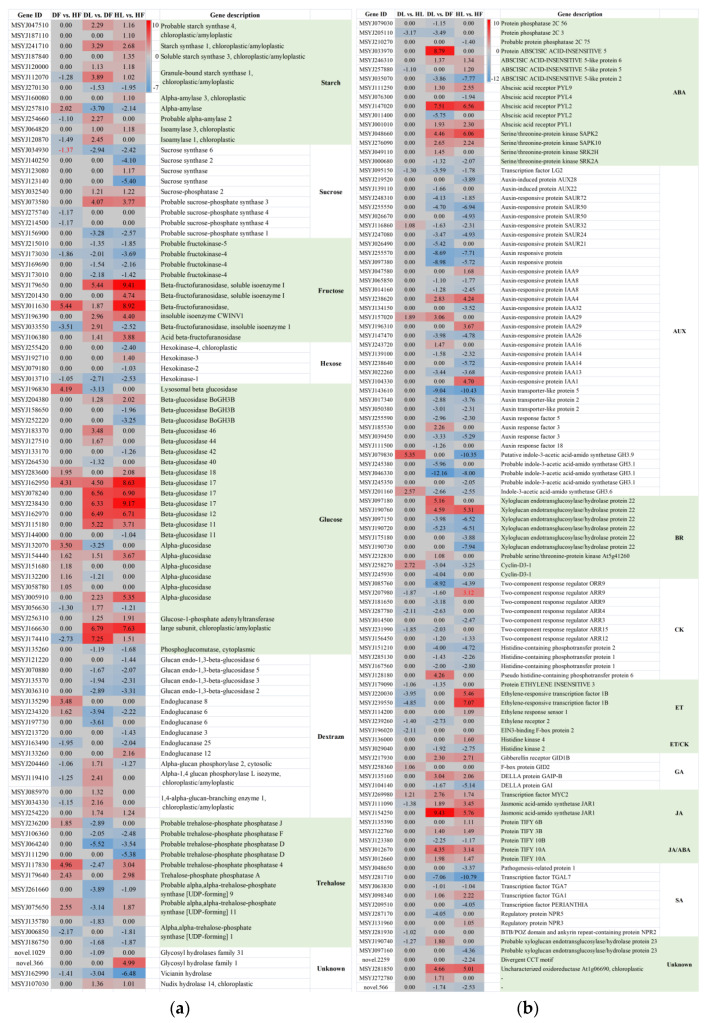
Log_2_Fold change cluster diagram of the differentially expressed genes in the starch and sucrose metabolism pathway (**a**) and plant hormone signal pathway (**b**). DF, diseased fruit pith. DL, diseased leaf. HF, healthy fruit pith. HL, healthy leaf. The color bar represents a range of log_2_Fold change values.

**Figure 8 ijms-23-15648-f008:**
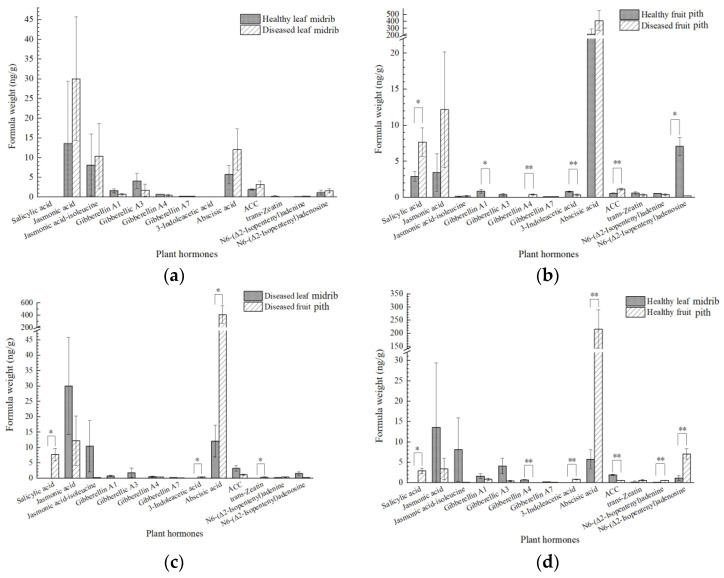
Endogenous phytohormone content in “*Candidatus* Liberibacter asiaticus” infected or uninfected leaf midribs and fruit piths. (**a**) Endogenous phytohormone content in CLas-infected and -uninfected leaf midribes. (**b**) Endogenous phytohormone content in CLas-infected and -uninfected fruit piths. (**c**) Endogenous phytohormone content in CLas-infected leaf midribes and fruit piths. (**d**) Endogenous phytohormone content in healthy leaf midribes and fruit piths. Data are represented as mean ± standard deviation. ACC, 1-aminocyclopropanecarboxylic acid. Means marked with asterisks are significantly different from corresponding control values (LSD test, * indicates *p* < 0.05, ** indicates *p* < 0.01).

**Figure 9 ijms-23-15648-f009:**
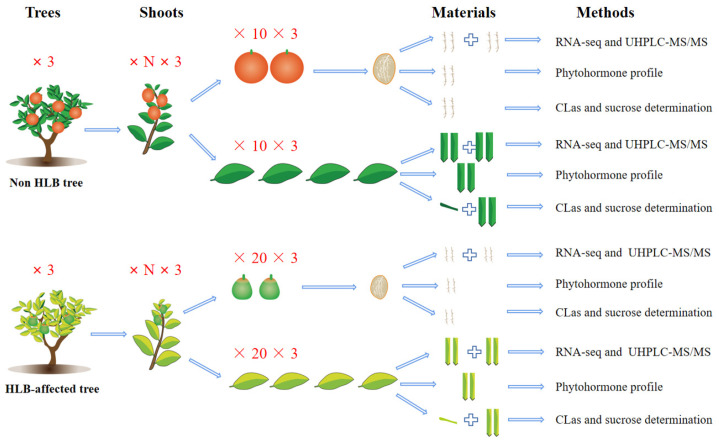
Diagram of the main materials and methods in this study.

## Data Availability

All transcriptomic data are available at NCBI with the accession number PRJNA883800.

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
