# Peer review of "Integrated Transcriptome and Metabolome Analysis Reveals Phenylpropanoid Biosynthesis and Phytohormone Signaling Contribute to “Candidatus Liberibacter asiaticus” Accumulation in Citrus Fruit Piths (Fluffy Albedo)"

_ijms, 2022, doi:10.3390/ijms232415648_

Round 1

Reviewer 1 Report (Previous Reviewer 1)

The authors have addressed all the issues I raised, I think it is now suitable for accept and publish on IJMS in the present form.

Author Response

Dear  reviewer,

we are really very grateful to you for your positive evaluation on the manuscript and thanks for taking your time to handdle this manuscript.

Sincerely,

Meirong Xu 

Reviewer 2 Report (Previous Reviewer 2)

The manuscript is improved a lot after revision. The authors apply transcriptome and metabolome analysis to explore why CLas is accumulated in the fruit piths and the differences in defense responses in plant leaves and fruits. This manuscript is informative, with diverse methods deployed.

Main comment: the authors need to provide more evidence to claim their opinions in the discussion part. All your conclusions based on the analysis results need to be more convincing (see comments below). Please make sure all the cited references are relevant to your research and explain the reference that you choose more clearly.

Other comments:

Line 203, for section 2.2. Quality control of RNA-seq and UHPLC-MS/MS. I know you explained detailed information in the method part. Please also briefly describe the experiment design for transcriptome and metabolome analysis here, such as how and when you collect the samples.

Line 346, please provide references for the biomarkers of CLas infection.

Line 374, I suggest revising the subtitle of section 2.5, which needs to be clarified and appropriate with your description.

Line 376-378, Do you mean the photosynthesis pathway is induced in healthy leaves and diseased fruits? Please give a concise description.

Line 436, Which aspect of tissue differences? Do you mean tissue difference in CLas accumulation or tissue difference in hormone levels? Please give a more accurate description.

Line 430-433 and others throughout the manuscript, please don’t overuse “vs” a lot. You could describe this kind of comparison with “vs” in your figure and table but try to explain more clearly in your text. You could apply “than, compared to, relative…”

Line 584-585, please provide the figure number.

Line 596-599, you mentioned previously hormone signaling-related DEGs in DL vs DF (78) and HL vs HF (79); why do you describe 53 of the 54 common DEGs in the following sentence? I think it should be some of 78 and some of 79… I am confused, and what is “common” DEGs mean.

Line 599, Why this confirmed the reliability of your transcriptome assay? Do you have any standards of reliability? Please provide evidence.

Line 601-602, Why the fold change data of ARR9 (MSYJ207980) in your transcriptome analysis is incorrect? Because its opposite expression in two groups? Does it mean that your data is not reliable? Why do you point this out? Did you conduct at least three independent experiments to collect transcriptome and metabolome analysis samples? I mean, at least three biological repeats. If you only do the experiment once and collect samples, the fold change data related to ARR9 (MSYJ207980) might not be representative.

Line 600, what is a “two-component response”? Please explain it in your manuscript.

line 623, line 625, what are the stress responses? Biotic stress? What pathogen? Is this stress response similar to CLas infection?

Here and in other cases in your manuscript, please make sure all the cited references are relevant to your research and explain the reference that you choose more clearly. In addition, please provide the connection between the reference and your study. Otherwise, your conclusions are not supported by the results. For example, it is not good enough to just present your data with JA and say that JA is important. It is better to explain why JA is important in your study (CLas infection in citrus). CLas is transmitted by the citrus psyllid, and JA plays a critical role in necrotrophic pathogen infection and herbivore attacks. So, there might be a close connection between JA and CLas accumulation. However, I didn’t find any explanation of the role of JA in your specific case. The same issue is in discussion sections 3.2, 3.3, and 3.4.

Line 652-653, how do shikimate O-hydroxycinnamoyltransferase and phenylalanine ammonia-lyase contribute to plants’ biotic and abiotic response of plants? Please provide more comments.

Author Response

The manuscript is improved a lot after revision. The authors apply transcriptome and metabolome analysis to explore why CLas is accumulated in the fruit piths and the differences in defense responses in plant leaves and fruits. This manuscript is informative, with diverse methods deployed.

Main comment: the authors need to provide more evidence to claim their opinions in the discussion part. All your conclusions based on the analysis results need to be more convincing (see comments below). Please make sure all the cited references are relevant to your research and explain the reference that you choose more clearly.

Response: Thanks very much for taking your time to review this manuscript. I really appreciate all your comments and suggestions! We deleted some uncertain conclusions or discussions, explained more of the references in some places of discussion when necessary. We have also double-checked all the references and have made sure they were cited correctly. For others, please find my itemized responses below and my revisions/corrections in the renewed files.

Other comments:

Line 203, for section 2.2. Quality control of RNA-seq and UHPLC-MS/MS. I know you explained detailed information in the method part. Please also briefly describe the experiment design for transcriptome and metabolome analysis here, such as how and when you collect the samples.

Response: Accepted. We added a sentence about the materials in the beginning of the section 2.2.

Line 346, please provide references for the biomarkers of CLas infection.

Response: We deleted the sentence.

Line 374, I suggest revising the subtitle of section 2.5, which needs to be clarified and appropriate with your description.

Response: Accepted. We revised it as: Photosynthesis and starch and sucrose metabolism were related to the citrus responded to CLas

Line 376-378, Do you mean the photosynthesis pathway is induced in healthy leaves and diseased fruits? Please give a concise description.

Response: Actually, the photosynthesis pathway is induced in leaves and diseased fruits. We revised the sentence as: The photosynthesis pathway was significantly induced in the leaves (HL vs HF and DL vs DF) and in the diseased fruits (DF vs HF). (Lines 366-367)

Line 436, Which aspect of tissue differences? Do you mean tissue difference in CLas accumulation or tissue difference in hormone levels? Please give a more accurate description.

Response: Accepted. We revised it as: This suggested that the enriched phytohormone signal transduction pathway was mainly related to the differences of tissue parts. (Lines 430-431)

Line 430-433 and others throughout the manuscript, please don’t overuse “vs” a lot. You could describe this kind of comparison with “vs” in your figure and table but try to explain more clearly in your text. You could apply “than, compared to, relative…”

Response: Accepted and revised accordingly.

Line 584-585, please provide the figure number.

Response: Accepted. “(Figure 7b)” was added in the text.

Line 596-599, you mentioned previously hormone signaling-related DEGs in DL vs DF (78) and HL vs HF (79); why do you describe 53 of the 54 common DEGs in the following sentence? I think it should be some of 78 and some of 79… I am confused, and what is “common” DEGs mean.

Response: We revised this section as: Totally 54 plant hormone signaling transduction pathway-related DEGs were common in DL vs DF and HL vs HF (Figure 7b). We further found that 53 of the 54 DEGs differed identically (35 were down-regulated in both groups and 18 were up-regulated in both groups). (Lines 587-590)

Line 599, Why this confirmed the reliability of your transcriptome assay? Do you have any standards of reliability? Please provide evidence.

Response: We add a sentence. “This confirmed the reliability of our transcriptome assay, as both groups (DL vs DF and HL vs HF) indicated the DEGs in leaves comparing to the fruit piths.” (Lines 591-592)

Line 601-602, Why the fold change data of ARR9 (MSYJ207980) in your transcriptome analysis is incorrect? Because its opposite expression in two groups? Does it mean that your data is not reliable? Why do you point this out? Did you conduct at least three independent experiments to collect transcriptome and metabolome analysis samples? I mean, at least three biological repeats. If you only do the experiment once and collect samples, the fold change data related to ARR9 (MSYJ207980) might not be representative.

Response: Yes, we conducted three independent experiments for transcriptome and metabolome analysis. There are 54 common DEGs in DL vs DF and HL vs HF, 35 were down-regulated in both groups and 18 were up-regulated in both groups. Only MSYJ207980 (ARR9) has the opposite value in the two comparison. Besides, there are 7 DEGs with annotation of “two-component response regulators”. Six of them were down-regulated leaves (DL vs DF and/or HL vs HF). However, the only one, MSYJ207980 (ARR9), was up-regulated (with Log2foldchange of 3.12) in HL vs HF. Based on these, we speculate the value of 3.12 is incorrect. So we wrote “we inferred that the fold-change data of this gene in HL vs HF might be incorrect”. But as you said, the experiment should be reliable if we had conducted 3 biological repeats. So we delete the section, as it is independent with other contents of the paragraph.

Line 600, what is a “two-component response”? Please explain it in your manuscript.

Response: Accepted. A reference was added here.

line 623, line 625, what are the stress responses? Biotic stress? What pathogen? Is this stress response similar to CLas infection?

Response: We revised the sentence as: The involved JAR1 was reported to function in sustaining flower development in Arabidopsis and and bacterial blight response in rice [55,56]. (Lines 621-622)

Here and in other cases in your manuscript, please make sure all the cited references are relevant to your research and explain the reference that you choose more clearly. In addition, please provide the connection between the reference and your study. Otherwise, your conclusions are not supported by the results. For example, it is not good enough to just present your data with JA and say that JA is important. It is better to explain why JA is important in your study (CLas infection in citrus). CLas is transmitted by the citrus psyllid, and JA plays a critical role in necrotrophic pathogen infection and herbivore attacks. So, there might be a close connection between JA and CLas accumulation. However, I didn’t find any explanation of the role of JA in your specific case. The same issue is in discussion sections 3.2, 3.3, and 3.4.

Response:  Accepted. Considering your suggestion, we have read through the discussion part and try our best to optimize it.

Line 652-653, how do shikimate O-hydroxycinnamoyltransferase and phenylalanine ammonia-lyase contribute to plants’ biotic and abiotic response of plants? Please provide more comments.

Response: We revised the section as: The ET induced PAL activity and PAL mRNA were induced by chilling injury in citrus cultivars showing chilling damage, indicating that PAL stands as a defense response of citrus against chilling stress [66]. HCTs were reported to be associated with lignin biosynthesis, involved in defense against abiotic and biotic stress [67].(Lines 649-652)

Reviewer 3 Report (New Reviewer)

The manuscript provides significant information about phytohormone signaling in HLB-infected citrus fruit pith and its comparison with leaves. The authors did a great job in bundling all the data, however, the introduction needs a lot of improvement. Most of it is a discussion on other transcriptome and metabolomic papers. The introduction should include a rationale to carry out the present experiment in terms of the current situation of HLB. I also suggest the authors do a validation of the gene expressions for 4 or 5 DEGs.

Line 52. Is mottling

Line 82. …, and they found…

Line 90. Start with the relevance of metabolomics in HLB. There is no flow in writing.

Line 193. Remove obviously.

Author Response

The manuscript provides significant information about phytohormone signaling in HLB-infected citrus fruit pith and its comparison with leaves. The authors did a great job in bundling all the data, however, the introduction needs a lot of improvement. Most of it is a discussion on other transcriptome and metabolomic papers. The introduction should include a rationale to carry out the present experiment in terms of the current situation of HLB. I also suggest the authors do a validation of the gene expressions for 4 or 5 DEGs.

Response: Thanks very much for your kind comments and suggestions. We have revised the introduction part according your comments. In order to connect the previous related studies and our study, the modifications were mostly done at the ends of some paragraphs. To confirm the reliability of trancriptome data, the expression level of 10 unigenes involved in phenylpropanoid biosynthesis, phytohormone signaling transduction, starch and sucrose metabolism, and photosynthesis were detected using real-time quantitative PCR. The results (not showed in the manuscript) showed the expression lebel of them were accorded with the transcriptome data.

Line 52. Is mottling

Response: Accepted.

Line 82. …, and they found…

Response: Accepted, thanks.

Line 90. Start with the relevance of metabolomics in HLB. There is no flow in writing.

Response: Thank you for pointing out this. We have revised the paragraph.

Line 193. Remove obviously.

Response: Accepted, thanks.

Reviewer 4 Report (New Reviewer)

Dear Editors,

Thank you so much to choose me as a reviewer of the manuscript IJMS (ISSN 1422-0067) entitled “Integrated transcriptome and metabolome analysis reveals phenylpropanoid biosynthesis and phytohormone signaling contribute to “Candidatus Liberibacter asiaticus” accumulation in citrus fruit piths (fluffy albedo). I hope that my comments will help Authors to improve their manuscript.

Detailed remarks concerning the manuscript.

The clear purpose and scientific hypothesis of the report should be presented together with the answer to the question stated as scientific hypothesis.

I suggest to give the clear practical aspects of the studies

The clear conclusions and the directions for the future studies should be given.

Key words: It is not recommended to use as key words the words or phrases that appeared in the title of the manuscript. Please do needed changes.

Author Response

Detailed remarks concerning the manuscript.

The clear purpose and scientific hypothesis of the report should be presented together with the answer to the question stated as scientific hypothesis.

Response: Thank you for your valuable comment. This has also suggested by other reviewers above. We have tried our best to improve and made some changes in the manuscript. These changes can be tracked in the modified manuscript.

I suggest to give the clear practical aspects of the studies

Response: We are sorry that we may have not expressed it clearly. We have done it according to your ideas.

The clear conclusions and the directions for the future studies should be given.

Response: We agree with you. We have added some conclusions and pointed out the directions for the future studies in each part of results. For example, Lines 28-33, Lines 296-298, Lines 322-324, Lines 342-345, Lines 360-363, Lines 385-387, et. al.

Key words: It is not recommended to use as key words the words or phrases that appeared in the title of the manuscript. Please do needed changes.

Response: Accepted, thanks.  We changed the key wards as: Huanglongbing; sequencing; metabolic; titer; fruit

This manuscript is a resubmission of an earlier submission. The following is a list of the peer review reports and author responses from that submission.

Round 1

Reviewer 1 Report

This submitted manuscript Integration of transcriptome and metabolome reveals  phenylpropanoid biosynthesis and phytohormone signaling contribute to “Candidatus Liberibacter asiaticus” accumulation in the citrus fruit piths (fluffy albedo)” by Cui et al., describes the influence of Huanglongbing on two Candidatus Liberibacter asiaticus”-accumulated citrus tissues. The authors show several phenotypes of HLB on these two parts. Based on the phenotyping, transcriptome and metabolome characterized DEGs and DAMs indicated the phenylpropanoid biosynthesis, phytohormone signaling transduction, starch and sucrose metabolism, and photosynthesis pathways were strongly contributed to the differences of tissue responses to CLas.

Overall, the experiments are well designed and conducted, and most results are clearly described, though a few of them few should be double checked. The conclutions are interesting, though not concise enough. There are a number of minor typographical errors and suggested alternate wording are provided.

Here are my suggestions to improve this manuscript:

Major:

To make the study achieve the stated aims, results of the later two experiments should explain the phenotypes like symptoms, CLas titers, starch or sucrose concentrations. Besides, we suggest the discussions are also .

Minor:

Line 137, change “high concentration of CLas” to “high titer of CLas”, also in Line 477 and Line 592.

The abbreviations, including SA, IAA DF, HF, DL, HL, need to be described at the first appearance and only once right after the first appearance in the manuscript.

The manuscript needs a thorough grammar and spelling check.

In the second paragraph of 2.6, please mention quantitative differences of phytohormones rather than simply “higher” or “lower”. This thing needs to be corrected throughout the manuscript about the DEGs and DAMs.

Line 95   The phraslead to alterations” in may be wordy, consider changing the wording.  

 lead to alterations--alter

Line100-101  Dangling modifier. The subordinate phrase “Referring to different tissues of citrus” does not appear to be modifying the subject it. Rewrite the sentence to avoid a dangling modifier

Line 108   Spelling corrections are needed.  “metabome”-- “metablome”

Line 124-126   The sentence may be unclear . Consider rephrasing.

Several categories of hormones are known to be produced in plants and defend plants by promoting specific protective mechanisms against biotic stresses

Line 129-130   The phrase “give full play” to may be wordy, consider changing the wording.

--play fully

Line136   It seems that “conclusions” may not agree in number with other words in this phrase.  

--conclusion

Line 137   The sentence may be unclear or hard to follow. Consider rephrasing.

A significantly high concentration of CLas was detected in fruit piths compared with leaf midribs.

Line 156   Please correct the” tense”    were-was

Line 160   Spelling correction is needed      verus --versus

Line 258   It seems that “CC related” is missing a hyphen.

Line 284-286  The sentence may be unclear .  

Three of the four were enriched by three pathways, namely photosynthesis, plant hormone signal transduction, and starch and sucrose metabolism.

Line 438-440  Hard-to-read sentence

Line 449  Spelling corrections are needed.

In the phenylpropaninoid--In the phenylpropanoid

Line470  The comma may be separating the subject and verb in the sentence.

Line 511  The phrase “the dynamics of starch “may be wordy.

--Starch dynamics .

Line576-578  Inconsistent punctuation.You used two different styles of apostrophes in your document .both styles are acceptable, but it’s best to be consistent .

Line 585   Unclear antecedent, It may be unclear who or what “This” refers to.

Line599-601  Consider rewriting this sentence in the active voice and specifying who or what performed the action. The passive voice isn't an error but it may be less clear and compelling in general writing.

Line 624  It seems that preposition use may be incorrect here.

--Figure 9.  Diagram illustration of the the main materials and methods of this study.

Line 630  Grammar corrections are needed.  --measured

Line 660  The word “asses” doesn’t seem to fit in this context   --assess

Line 705  Grammar corrections are needed  --was added to a cold...

Line 712  Spelling corrections are needed.  --control homeostasis.

Besides, please double-check all the figures and tables.

Reviewer 2 Report

The authors confirmed the accumulation of CLas in piths of ripe ‘Shatangju’ fruits compared with leaf midribs and further explored this phenomenon’s biological basis via combined biomics. The authors tried integrating transcriptome and metabolome analysis to dig for more information, which is attractive to readers. And they found that phenylpropanoid biosynthesis, phytohormone signaling transduction, starch and sucrose metabolism, and photosynthesis might be involved in CLas accumulation in fruits. Although this manuscript is informative, I think the research is not novel as the phenomenon has already been discovered and studied. The figures/tables are fine. The presentation of data (Figures) showed detailed information. But the authors need to reorganize all the information from the figures to clarify the research topic, not only explain everything in those figures. Otherwise, you can’t provide sound scientific evidence. The readers will be misled and lost. Furthermore, authors would benefit from a language editor, as the text and sentence structure is confusing in many instances. For those reasons, I think the manuscript needs major modifications before further evaluation for publication.

Major comments:

Before further submission for publication, I strongly recommend that authors send the manuscript to be edited by a language reviewer. The structure of many sentences appears to be broken. Moreover, there exist a lot of grammar issues. For instance, lines 37-38, 41-42, 139-140, and 212-213 have grammar issues and are hard to read. I suggest revising these sentences to be clear and understandable.

Line 23-24, the authors might have to think carefully. CLas high accumulation led to differential gene expression in different tissues. DEGs are not the cause but an effect.

Line 120, line 129-130, line 130-131, add references

Figure 1(d) and line 171, what is the unit of Quantitation of CLas in leaf and fruit samples? Please provide the information in the manuscript. Figure 1 (e), what is the unit of sucrose content? Please provide the information in your manuscript

In Table S1 and Table S2, the sample names are unrelated to the note you put under the table. What is the meaning of BG, BY, JG and JY?

Line 184-186, could you please provide the figure number of the PCA analysis?

Line 199, please provide references for HMDB and Lipidmaps database

Please revise your figure legend in lines 203, 204, and 205 in figure 2 and be consistent with other figure legends. (a) is/(b) is/(c) is not appropriate and formal writing.

In Figures 3 and 4, you compared the DEGs in different groups. Have you specifically compared up-regulated genes and metabolites in other groups? Also, have you compared down-regulated genes and metabolites in different groups? Please revise your figure legend. Be formal writing and provide detailed information. Readers should be clear with all your figures.

L293, what are “four, 0, 10 and one pathways”? I can’t understand this phrase.

Please revise your figure legend in figure 5. Be formal writing. “(a) and (b) are…” is not appropriate. The up-regulated genes and metabolites in KEGG nodes were shown red/green. Genes and metabolites cannot be red/green. Revise your grammar.

Line 327, line 462, affection? Do you mean infection?

Line 328-331, difficult to understand your statement. Revise your sentences to be more precise and understandable.

Line 342-343, grammar issue. Depressed? Do you mean decreased or reduced?

Figure 6. What is the meaning of the color?

Line 356-359, hard to understand this sentence.

Line 366-370, problems with grammar and sentence structure. Hard to understand this sentence.

The same as Figure 6, what is the meaning of the color in Figure?

Line 376, the subtitle is not appropriate. It could be improved. For example, Plant hormone signal transduction was induced upon CLas infection in leaves.

For results 2.3-2.7, please explain the analysis results clearly and precisely, not explain everything in the figures. Focus on your research topics.

For the discussion part, I would suggest revising all the subtitles. The current ones couldn’t present the text that you discussed. Please insert the figure numbers in the discussion part for the convenience of readers (Line 486 and others). Please revise the grammar and sentence structures throughout the whole manuscript (for instance, line 507, induced by HLB infection, not induced by HLB-affected leaves, lines 543-547, and many others…)

Round 2

Reviewer 1 Report

The authors answered and solved all the issues I raised in version one. I think it is suitable for acceptance in its present form.

Reviewer 2 Report

Thanks for the point-to-point reply from the authors. The manuscript is improved after revision. But I think the manuscript needs a lot of work. The authors need to enhance their scientific writing skills. The introduction did not describe the references very clearly. For example, the authors only mentioned differences in tested groups but did not show what the difference was (higher or low? More or less?...). And they don’t explain why to point out some data from published papers (see comments below). Please read the references carefully and integrate them into your introduction clearly—the same issues with the results and discussion part.

This manuscript still requires extensive editing of the English language and style. The text is confusing, and the sentence structure is too crowded and repeatable in many instances. A lot of grammar issues!

And as I mentioned before, although this manuscript is informative, the research is not novel as the phenomenon has already been discovered and studied. Could the authors explain the novelty or difference from other papers in the discussion part?

1.     Line 26-28 in the revised version, I think the authors still need to make efforts to revise the sentence structure. Just take this sentence, for example. Notably, up-regulated beta-glucosidase and endoglucanase genes and down-regulated defense response genes or metabolites were identified in fruits.

Hard to read. A lot of phrases in one sentence. And they are not closely related. Why not separate them into two sentences? Notably, we discovered that gene expression related to beta-glucosidase and endoglucanase was up-regulated in fruits. And defense-related gene expression and metabolite accumulation were significantly downregulated in infected fruits. Please think about other sentences throughout your manuscript.

2.     Line 88-93, your description is “Fang et al. [8] analyzed the differences in leaves and fruit piths between healthy and infected CLas by interactive transcriptome analysis. Compared with leaves, fruit piths had higher CLas genome depth. Compared with fruit piths, the significant up-regulated host DEGs were involved in the biosynthesis of antimicrobial-associated secondary metabolites in leaf midribs.”

What is “between healthy and infected CLas”?? I know what you mean. But this kind of description is not appropriate for a paper to be published for readers.

And the last sentence, if you want to say, “compared with A,” the following sentence should be some difference between A and B. However, I didn’t find any words showing this comparison. Do you mean leaf midribs have more up-regulated DEGs than fruit piths, which are involved in the biosynthesis of antimicrobial-associated secondary metabolites?

If so, I would revise it like “Fang et al. apply transcriptome analysis to find the difference in leaves and fruit piths with and without CLas infection. The fruit piths had higher CLas genome depth than fruit piths. Compared with fruit piths, more upregulated genes involved in the biosynthesis of antimicrobial-associated secondary metabolites were identified in leaf midribs.”

3.     Line 99-103, your description is: Some experiments analyzed the leaves of sensitive and tolerant citrus through metabolomics and revealed that after the HLB-sensitive citrus was infected with CLas, the contents of L-threonine, L-serine and mannose in the leaves were significantly changed, the contents of L-proline, indole, aspartate and glutamate of disease-resistant citrus also changed significantly.

First of all, what are the references of this sentence? Please provide references.

Sensitive and tolerant citrus? Do you mean resistance and susceptible? And it might be more common to say, for example, Citrus cold-tolerant/-sensitive varieties or citrus sensitive/tolerant to cold stress.

And how were these contents changed in the tolerant/sensitive varieties? Up or down-regulated? If you decide to put a reference in your introduction, please clarify it concisely and clearly. And a lot of your sentence structure is too crowded and repeatable. It could be revised as some reports (reference) revealed that there existed extreme differences of metabolites in the leaves of HLB-sensitive and resistant citrus to CLas infection. The contents of L-threonine, L-serine and mannose in the leaves were… and/but the contents of L-proline, indole, aspartate and glutamate were…in HLB-resistant citrus.

4.     Line 103-104, your description is: Infected-CLas citrus can also alter the metabolism of long-chain fatty acid. It is more appropriate to use passive voice in scientific works.

Infected-CLas citrus? I don’t understand!

5.     Line 105-109, your description is: Furthermore, different concentrations of several metabolites, such as phenylalanine, histidine, limonin and synephrine, were found in diseased and healthy fruits [24]. There were also significant differences in the concentrations of sugars, amino acids, organic acids and limonin glucoside between orange juice collected from CLas-infected citrus trees and healthy trees [25].

The same issue is mentioned above. If you decide to put a reference in your introduction, please clarify it concisely and clearly. What are the different concentrations? Higher or lower? In diseased or healthy fruits? Without infection or after infection?

6.     Line 109-112, your description is: When comparing different tissues of citrus, we found that the sucrose content in infected leaves was higher, but the sucrose content in roots did not change significantly and a small amount of proline, betaine and malate were detected in HLB-affected leaves [26].

First of all, this is other researchers’ work, not yours. So, not “we found”.

Second, the sentences are crowed and repeatable.

Moreover, please explain a little bit about the reference, not only showing the data. Please let us know why you specifically describe this reference and related points in your introduction. So what is the meaning of “a small amount of proline, betaine and malate were detected in HLB-affected leaves”? Are these compounds related to defense or others?

Please consider revising: Comparing different tissues of citrus after CLas infection, it was found that the sucrose content was increased in leaves but not roots. And a small amount of proline, betaine and malate were detected in HLB-affected leaves, which is….

A lot of similar problems in the manuscript. I think your manuscript needs a lot of work. I would suggest rewriting every part and letting your peers or collaborators revise it.
